# Antibacterial-Agent-Immobilized Gelatin Hydrogel as a 3D Scaffold for Natural and Bioengineered Tissues

**DOI:** 10.3390/gels5020032

**Published:** 2019-06-11

**Authors:** Tuyajargal Iimaa, Takaaki Hirayama, Nana Shirakigawa, Daisuke Imai, Takanobu Yamao, Yo-ichi Yamashita, Hideo Baba, Hiroyuki Ijima

**Affiliations:** 1Department of Chemical Engineering, Graduate School of Engineering, Kyushu University, 744 Motooka, Nishi-Ku, Fukuoka 819-0395, Japan; tuyajargal@mnums.edu.mn (T.I.); t.r.y.k.t.t@kyudai.jp (T.H.); shirakigawa@chem-eng.kyushu-u.ac.jp (N.S.); 2Department of Biochemistry and Laboratory Medicine, School of Biomedicine, Mongolian National University of Medical Sciences, Ulaanbaatar 14210, Mongolia; 3Department of Surgery and Science, Graduate School of Medical Sciences, Kyushu University, 3-1-1, Maidashi, Higashi-ku, Fukuoka 812-8582, Japan; dice.m325@gmail.com (D.I.); y-yama@kumamoto-u.ac.jp (Y.-i.Y.); 4Department of Gastroenterological Surgery, Graduate School of Medical Sciences, Kumamoto University, 1-1-1 Honjo, Chuo-ku, Kumamoto 860-8556, Japan; i_am_kida7172205@yahoo.co.jp (T.Y.); hdobaba@kumamoto-u.ac.jp (H.B.)

**Keywords:** hydrogel, antibacterial agent, gelatin, 3D scaffold, biomaterials for tissue engineering, cytotoxicity, cell culture

## Abstract

Hydrogels and their medical applications in tissue engineering have been widely studied due to their three-dimensional network structure, biocompatibility, and cell adhesion. However, the development of an artificial bile duct to replace the recipient’s tissue is still desired. Some challenges remain in the tissue engineering field, such as infection due to residual artifacts. In other words, at present, there are no established technologies for bile duct reconstruction as strength and biocompatibility problems. Therefore, this study investigated hydrogel as an artificial bile duct base material that can replace tissue without any risk of infectious diseases. First, an antibacterial agent (ABA), Finibax (an ABA used for the clinical treatment of biliary tract infection), was immobilized in gelatin using a crosslinking agent, and the antibacterial properties of the gel and its sustainability were tested. Furthermore, the immobilized amount and the improvement of the proliferation of the human umbilical vein endothelial cells (HUVECs) were cultured as the ABA-Gelatin hydrogel was introduced to prepare a 3D scaffold. Finally, we performed hematoxylin and eosin (H&E) staining after subcutaneous implantation in the rat. Overall, the ABA-Gelatin hydrogel was found to be viable for use in hydrogel applications for tissue engineering due to its good bactericidal ability, cell adhesion, and proliferation, as well as having no cytotoxicity to cells.

## 1. Introduction

In general, tissue engineering and regenerative medicine require scaffolds that have a cell adhesion surface and cell growth area [1] and seek to invent desired parts for living organisms [2]. The importance of tissue engineering is demonstrated by the demand for tissue transplantation and regeneration [3]. The application of biopolymer-based hydrogels dates back to 1960 when researchers introduced the use of soft contact lens material [4]. Hydrogels possess flexibility very similar to natural tissue due to their large water content, soft consistency, and excellent biocompatibility [5]. They also show a minimal tendency to adsorb proteins from body fluids because of their low interfacial tension [4]. Therefore, in recent years, hydrogels have had many different functions in the field of tissue engineering. In addition, hydrogel scaffolds are being applied to transplant cells and to engineer tissues in the body, including cartilage, liver, and smooth muscle [2]. 

Biliary complications are a significant cause of morbidity and mortality following orthotopic liver transplantation, with estimates of their incidence ranging between 11.5% and 34%. In a survey of 30 major transplant centers in the US, biliary complications occurred in 16% of patients after liver transplantation [6]. Additionally, the post-operative course for an extra-hepatic bile duct is complicated by retrograde infections via the intestine or stenosis at the anastomosis requiring re-anastomosis [7]. There are some reports on an association of bacterial infection with the main disorders of the biliary tract [8]. For this reason, this study aimed to determine the effect of an antibacterial agent with gelatin gel on transplantation in a rat study. 

Finibax, as a 1-β-methyl carbapenem, is a cell-wall synthesis inhibitor. This carbapenem has been activated as both imipenem and meropenem against both gram-positive and gram-negative bacteria. Moreover, these kinds of antibiotic have excellent in vitro activity and pharmacological properties [9]. Particularly, they treat gram-negative bacteria that are resistant to most antibiotics, including carbapenems and cephalosporins. Surveillance of drug resistance has shown that bacteria-expressing β-lactamases have attracted the attention of modern researchers due to their resistance to a broad range of β-lactams, including third-generation cephalosporin derivatives [10]. Carbapenemases are β-lactamases with versatile hydrolytic capacities. Therefore, they can be used for both in vivo and in vitro biomedical applications [11]. 

As mentioned above, hydrogels have been utilized as scaffold materials for drug delivery and a variety of other applications [12]. This paper focuses on the use of hydrogel as a scaffold for tissues engineering. Therefore, we fabricated a bile duct scaffold using a natural polymer with an antibacterial agent and then transplanted the scaffold into rats in order to investigate the functional potential as a replacement for the native bile duct with its morphology after transplantation. We identified three categories of scaffold applications in this paper—the antibacterial agent, the bioactive molecule delivery, and the cell/tissue delivery. It is expected that this 3D scaffold for transplantation is useful for regenerative medicine based on tissue engineering. 

## 2. Results and Discussion

### 2.1. Selection of an ABA

In this section of the study, our aim was to develop a scaffolding base material with antibacterial properties, and three antibacterial agents (ABA) were considered as candidates for this study—Finibax, Flomoxef, and Biapenem—as carbapenems and cephalosporins show broad antibacterial activities (Table 1). These ABAs exert antibacterial properties by inhibiting bacterial cell-wall synthesis and lysing. They can bind and inactivate the protein-binding protein, an enzyme for cell-wall synthesis [13,14]. 

The ABA solutions were dropped (Figure 1a,b) into position on a dish and the bactericidal effect against *Escherichia coli* (*E. coli*) was confirmed with each ABA (Figure 1c,d,f,g). We selected *E. coli* as a gram-negative (aerobic and anaerobic growth) bacteria. The antibacterial effect of the agents was more pronounced against gram-negative bacteria than gram-positive organisms [9]. Importantly, in a study by Tajeddin et al. a significant association was observed between the colonization of *E. coli* and biliary tract diseases [8].

First of all, we evaluated what concentration range of ABA exhibits the antibacterial effect. The effects of Finibax, Flomoxef, and Biapenem with *E. coli* were determined at around 2.0 µg/mL, 8.0 µg/mL, and 16 µg/mL or more, respectively. This indicated that each ABA concentration showed different effects in several parts. 

For spectral analysis, the maximum absorption wavelengths of Finibax and Biapenem were found at approximately 300 nm, while Flomoxef was found at 260 nm (Figure 1e). Additionally, by using a calibration curve of absorbance and concentration at that peak, we found that it was possible to easily measure the concentration of the ABA in the sample.

The phenomena suggest that the higher concentration of each ABA apparently enhanced the antibacterial effect (Figure 1c,d,f,g). In other words, it would be possible to prepare a scaffolding base material with antibacterial properties if the ABA could be contained at a concentration higher than these sufficient values.

The antibacterial activity results clearly showed that this study should use Finibax, which showed its effect at a low concentration and had a large peak wavelength. In addition, hydrogels that contain β-lactam drug inhibitors, such as carbapenems, e.g., doripenem (Finibax), and cephalosporins, e.g., cefotaxime and ceftazidime [15], are used in urinary catheters, can prevent bacterial colonization on the surface, and provide a smooth and slippery surface to improve its biocompatibility [16]. Additionally, we considered the fact that Finibax is actually used for treatment by interfering in the synthesis of the bacterial cell-wall [15].

### 2.2. Immobilization of the ABA

We tried to immobilize the ABA as antibacterial sustainability can be improved by crosslinking an ABA with gelatin. The hydrogel removed from a 48-well plate is shown in Figure 2a. The crosslinking was carried out using 1-Ethyl-3-(3-dimethylaminopropyl)-carbodiimide (EDC)/*N*-hydroxysuccinimide (NHS). This agent reacted with the ABA and gelatin, which both have functional groups. β-lactams, as shown in Figure 2b, have a four-membered lactam ring [15], while gelatin is a heterogeneous mixture of single- or multi-stranded polypeptides, each with extended left-handed proline helix conformations and contain between 50 and 1000 amino acids [17].

Hydrogels do not disintegrate during swelling, thanks to their crosslinked structure [16,18]. The chemical crosslinking covered here involved the grafting of monomers on the backbone of the polymers or the use of a crosslinking agent to link two polymer chains through their functional groups (such as OH, COOH, and NH_2_) with a crosslinker, such as EDC [19]. EDC is a zero-length crosslinking agent used to couple carboxyl or phosphate groups to primary amines. 

To increase the stability of this active ester, NHS or *N*-hydroxysulfoxuccinimide (sulfo-NHS) can be used. One of the main advantages of EDC/NHS coupling is water solubility, which allows direct bioconjugation to be carried out under physiological conditions [20]. In addition, using the EDC/NHS coupling, almost all kinds of molecules (e.g., enzymes, antibodies, peptides, DNA, and fluorophores) can be attached to the surface without prior modification [21].

Figure 2c shows the ratio of the released ABA from the gels using a calibration curve. About 90% and 50% of the ABA was released slowly in the ABA-mixed gelatin gel and the ABA-immobilized gelatin gel, respectively. This result suggested that the ABA was successfully immobilized in gelatin by the EDC/NHS. From the release study, we found that the ABA treated with the gel was immobilized as well as released slowly. 

Next, in order to measure the amount of immobilized ABA, an evaluation was made by decomposing the gel. From the calibration curve, the ABA concentration of the decomposing solution was detected—the results are shown in Figure 2d. By comparing the mixing condition with the gelatin only, the equivalent results were obtained. It was assumed that the unfixed ABA was washed out. While, in the immobilized condition, the concentration was significantly greater than that of the other two conditions. It was confirmed as well that the ABA was immobilized by the EDC/NHS coupling. 

Additionally, we compared the crosslinking potential of the EDC/NHS couples in different molar ratios (Table 2). Our experimental results showed that the immobilized density in the gel could be manipulated by a crosslinking agent—the higher concentration of the EDC/NHS apparently enhanced the immobilized density of the ABA (Figure 2e). From this result, we focused on the second component that contains 20 mg/mL EDC:12 mg/mL NHS for future studies. This coupling can be immobilized, 250 µg/mL ABA is the effective value for Finibax (Figure 1d). 

### 2.3. Evaluation of the Antibacterial Activity of the ABA after Immobilization

In the previous experiment result, it was suggested that the ABA was immobilized to the gelatin gel. However, it was necessary to evaluate the antibacterial property of the ABA after immobilization.

*E. coli* activity was measured by the appearance of turbidity. Turbidity is generally used as an indicator of the number of bacteria [22]. Therefore, the correlation was investigated from the calibration curve of *E. coli*. The response of *E. coli* liquid was found to be linear in the investigated concentration range as shown in Figure 3a. The correlation coefficient was represented by the linear association between the bacteria number and the turbidity. It can be seen from the test results that the calibration curves for *E. coli* were correlative within a range of CFU/mL. From the counting result, the number of coliform bacteria in the *E. coli* liquid was calculated. The correlation between the number of *E. coli* and the turbidity was obtained. It was shown that the turbidity increased as the number of *E. coli* increased. 

As an antibacterial property of the solution, the turbidity was investigated for 3 h by mixing the *E. coli* liquid and the ABA solution in different concentrations. The changes in the turbidity over time at each ABA concentration are shown in Figure 3b. The turbidity results had the same values at all concentrations within 1 h. It was clear that after 1 h of mixing, the ABA concentrations were affected differently. Therefore, after 3 h, it was confirmed that the turbidity became lower as the concentration of the ABA increased. Our experimental results showed that the antibacterial activity improved as the concentration became higher. 

Furthermore, to investigate the antibacterial properties of the unknown specimen, the objective of this study was to prepare a calibration curve (Figure 3c). On the contrary, the result showed a good correlation for *E. coli*, with a correlation coefficient (r^2^) of approximately 0.88, showing a high correlation. Mainly, it was suggested that the antibacterial property of unknown specimens can be measured using the calibration curve. 

The amide bond is formed then between the ABA and the gelatin with crosslinking, the structure of the ABA is changed, and then the ABA may be inactivated. Therefore, to determine the activity of the ABA, we cultured *E. coli* in gel digestion solution. The ABA-free, the ABA without the immobilization, and the ABA with the immobilization were mixed with the *E. coli* liquid and cultured (Table 3). After 3 h of the culture, the turbidity (an indicator of the number of bacteria) was examined and the activities of the ABA were compared. The turbidity after 3 h of incubation is shown in Figure 3d. From the previous study, it was confirmed that the turbidity after 3 h of *E. coli* culture depended on the ABA concentration. The decomposed solution of the ABA-immobilized gelatin gel was mixed with *E. coli*, and the turbidity was measured at about 0.4. Similarly, the turbidity of ABA mixed with *E. coli* also showed a value of about 0.4. On the other hand, the same turbidity was detected in these solutions by spectral measurement. It was suggested that the gel digestion liquid had an equivalent antibacterial activity to the ABA concentration. Overall, after immobilization, the activity of the ABA was the same as before immobilization. 

### 2.4. Antibacterial Activity of the Gel 

To observe the activity of the ABA on the *E. coli* growth, the ABA-free gelatin gel and the ABA-mixed gelatin gel were placed on agar medium, the results are the images in Figure 4a,b. One day after the inoculation with *E. coli*, the whole agar medium appeared yellow. However, the agar medium became transparent in the ABA-mixed gelatin gel, and the growth suppression of *E. coli* was confirmed. It was visually confirmed that the ABA was gradually released from the gel and that the *E. coli* growth decreased. 

In addition, the number of bacteria on the gel surface 1 day after the culture was calculated from the number of colonies (Figure 4c,d). The doubling time was calculated from the number of bacteria at sowing: PBS was 89 min, while the LB medium was 62 min. In the LB medium, the doubling time was shorter because of the abundance of amino acids and minerals. Colonies were formed in both the PBS and LB mediums and were solvent under the condition without ABA. The bacterial number on the gel surface increased from 539 (on day 0) to 2.32 × 10^6^ in PBS and 1.01 × 10^8^ in LB medium (on day 1). In contrast, under the ABA immobilization conditions, the formation of colonies was not confirmed. This suggests the antimicrobial property of the ABA-immobilized gelatin means that bacteria do not grow on the surface upon transplantation and infection is prevented.

The antimicrobial property of the ABA-immobilized gelatin was confirmed in vitro. However, sometimes a surgical site infection occurs after surgery in vivo. For this reason, the gels were transplanted subcutaneously, the bacteria adhering to the gels was investigated, and the antibacterial property was evaluated in vivo. We measured the number of bacteria in rat subcutaneous implantation. No colonies of bacteria were confirmed in both the ABA-free and ABA-immobilized conditions (Figure 4e,f). This result suggested that there were no subcutaneous bacteria, so antibacterial activity under the skin could not be evaluated. On the other hand, in a clinical investigation, it is necessary to transplant into the abdominal cavity where there are organs, such as the intestines, and evaluate antibacterial activity.

Moreover, turbidity measurements were used to determine the increase in the ratio of turbidity for different gels. The growth ratio was calculated by dividing the turbidity and the initial turbidity (Figure 5a). In the ABA-free gel condition, the turbidity increased from the first time, and thereafter it hardly changed. The difference between the first and the second case was considered to be due to the first decrease in the turbidity rate by the EDC/NHS remaining in the gel. In contrast, under the immobilization condition, the turbidity did not increase at all until the third time, while it increased greatly after the fourth time. A possible reason for this increase is that the ABA, which had not been immobilized flowed out due to the gel being used without washing and the effect was exhibited. Although it increased after the fourth time, the turbidity rate was lower than in the ABA-free condition, and the growth of the bacteria could be inhibited. From the above, it is suggested that antibacterial properties are observed near the ABA-immobilized gelatin gel. 

### 2.5. Evaluation of Cytotoxicity of the Gel Surface

The tissue-engineered artificial bile duct is a combination of three elements, namely a cell, a scaffold, and a growth factor, and is a substitute for living tissue [1,23]. In recent years, hydrogel scaffolds are broadly used for cell delivery and tissue development as they have a highly hydrated 3D network, cell adhesion, proliferation, and differentiation as well as good chemical and mechanical signals to cells [23]. Therefore, hydrogels can be strategized for different applications: (i) to achieve a sustained release with minimum toxicity to the tissues and (ii) to form a responsive surface that possesses antibacterial activity and a self-cleaning ability for long-term usage [24]. 

Therefore, it is desirable that the scaffold shows no toxicity to cells. In order to investigate how the ABA-immobilized gelatin gel cultured on cells, human umbilical vein endothelial cells (HUVECs), which are a kind of epithelial cells, were selected. By comparing the immobilizing ABA-gelatin condition with the ABA-free condition, almost the same degree of cell proliferation was confirmed, as shown in Figure 5b. In addition, mitochondrial activity was evaluated by WST-8 assay, and cell morphology was observed (Figure 6a–d). These results suggested that there was no cytotoxicity by immobilizing the ABA and that the cell proliferation was good. Generally, the hydrophilic surfaces possess low interfacial free energy. Therefore, when in contact with body fluids, hydrogels have good biocompatibility and a low tendency to adhere cells on the gel surface [19]. Furthermore, we performed a visual evaluation by hematoxylin and eosin (H&E) staining the gel after its subcutaneous implantation in rats. 

In order to investigate how the ABA affected the tissue, the ABA-immobilized gelatin gel was subcutaneously transplanted on the back of a rat, taken out 1 week later, and stained with the H&E staining. Photographs of the gel after extraction are shown in Figure 7a,d. One week after the transplantation (Figure 7b,e), the gel was taken out of the tissue and was expected to have biocompatibility. Results of the H&E staining are shown in Figure 7c,f. The results of the H&E staining showed that cells were adhering on the surface of the gels under both conditions, and the gelatin gel was expected to be biocompatible. In addition, some of the color was thickened with an ABA-immobilized gelatin gel, and mild inflammation occurred. However, mild inflammation secretes various cytokines in vivo, including substances that promote tissue replacement. As a clinical study, it is necessary to evaluate this by conducting a long-term transplantation. 

Consequently, many materials have been used for one vital requirement as cells can easily be mixed with the gel. We were able to immobilize the ABA in the gelatin hydrogel for this reason. In conclusion, the ABA-immobilized gelatin hydrogel showed good antibacterial activity and low cytotoxicity both in vitro and in vivo.

## 3. Materials and Methods 

### 3.1. Chemicals and Reagents 

The materials and reagents in this study were obtained from the following sources: Gelatin from porcine skin (gel strength 300 Type A) was purchased from Sigma Aldrich Ltd (St. Louis, MO, USA); Doripenem and Flumarin were purchased from Shionogi & Co., Ltd (Osaka, Japan); Omegacin, Streptomycin sulfate, and Penicillin G potassium were purchased from Meiji Seika Pharma Cooperation Ltd (Tokyo, Japan); human umbilical vein endothelial cells (HUVECs) were obtained from Kurabo Industries Ltd (Osaka, Japan); collagenase, 2-(*N*-morpholino)ethanesulfonic acid (MES), D (+) glucose, agar, *N*-hydroxysuccinimide, HEPES, and 10% formalin were purchased from Wako Pure Chemical Cooperation (Osaka, Japan); Bacto tryptone, a pancreatic digest of casein, was purchased from Beston, Dickinson, and Company (Durham, NC, USA); Bacto yeast was purchased from Sigma Aldrich Co., Ltd (St. Louis, MO, USA); Fasting Blood Sugar (FBS) was purchased from Capricorn Scientific GmbH (Ebsdorfergrund, Germany); EGM-2 was from Lonza Walkersville MD (Walkersvile, MD, USA), and the WST-8 assay kit was obtained from Dojindo (Kumamoto, Japan). All other chemicals were of an analytical grade. Nine-week-old SD rats were obtained from SLC., Co., Ltd (Fukuoka, Japan). 

### 3.2. Selection of the Antibacterial Agent 

In the evaluation method, *E. coli* was seeded on an agar medium and the ABA solution at each concentration was dropped onto it. The concentration of the ABA in the solution was measured by the absorbance, then the peak wavelength was determined. The ABAs that were selected in this study are shown in Table 1.

#### 3.2.1. Evaluation of Effective Concentration

ABA aqueous solutions of 2–500 µg/mL were prepared. Agar medium (20 mL) was poured into 90 mm dishes, and allowed to stand at room temperature until solidified. *E. coli* was pre-cultured in a shaker at 37 °C for 1 day. This *E. coli* was spread on the agar medium. Thereafter, 10 μL of each ABA solution was added drop by drop. Cultivation was carried out in an incubator at 37 °C for 1 day, and the degree of proliferation was observed.

#### 3.2.2. Spectrum Measurement 

A 50 µg/mL aqueous solution of each ABA—Finibax, Flomoxef, and Biapenem—was prepared, and the absorbance of 100 µL of the solutions was measured using an ultraviolet-visible light spectrophotometer (UV-2500 PC, Shimadzu Corporation, Kyoto, Japan). 

### 3.3. Immobilization of the ABA

#### 3.3.1. Evaluation Based on the Release of the ABA into the Liquid

The solution was obtained from the mixture containing 200 mg/mL of gelatin and 10 mg/mL of ABA in a ratio of 1:1. The mixture was poured into a 48-well plate at 200 μL/well, and the gel was obtained after cooling at 4 °C for 15 min. Next, 200 μL/well of EDC/NHS (20 mg/mL:12 mg/mL) solution was dropped. It was allowed to stand at 4 °C for 1 h. After the preparation process, the gel was transferred to a 12-well plate, and 2 mL/well of ultrapure water was added. The absorbance of 100 μL of this solution was measured at 300 nm. From the calibration curve of the absorbance and concentration of each ABA, the amount of ABA that flowed out into ultrapure water was calculated by the following formula:
(1)ABA immobilized amount=the ABA concentration in decomposition solution×liquid volume

#### 3.3.2. Evaluation of the Immobilization amount by Gel Decomposition

The preparation method of the gels was as described in Section 3.3.1. The gels were then transferred to 50 mL centrifuge tubes. An amount of 20 mL of 0.05 M of 2-(*N*-morpholino)ethanesulfonic acid (MES) buffer was added drop by drop and gently shaken for 24 h. After the supernatants were removed, the same procedure was repeated with 20 mL of 5 M of NaCl, and 20 mL of 0.05 M MES buffer. After washing, the gels were transferred to a 12-well plate, and 2 mL/well of 0.05% collagenase solution was added. The solutions were gently shaken at 37 °C for 3 h. The absorbance of the decomposition solution was measured at 300 nm.

#### 3.3.3. Evaluation of the Crosslinking Potential with the EDC/NHS Couple 

The preparation method of the gels was as described in Section 3.3.1. Briefly, the ABA-Gelatin (10 mg/mL:200 mg/mL, 1:1) mixture was added to 200 μL of EDC/NHS solutions (Table 2) after cooling at 4 °C for 15 min. It was allowed to stand at 4 °C for 1 h and transferred to a 12-well plate. Then, 2 mL/well of ultrapure water was added. The absorbance was measured at 300 nm, and the amount of immobilized ABA was calculated. 

### 3.4. Evaluation of Antibacterial (AB) Activity of ABA after Immobilization 

#### 3.4.1. Correlation Evaluation of the E. coli and Turbidity

Frozen *E. coli* liquid was thawed at room temperature. The *E. coli* liquid and the LB medium were mixed at a ratio of 2:5. The mixture was shaken at 100 rpm in a water bath of 37 °C for 1 day. Each sample was diluted 100-fold, and the bacteria number was counted using a counter. 

#### 3.4.2. Correlation Evaluation of the ABA Concentration and Turbidity 

The mixture solution containing the *E. coli* liquid and the LB medium was described previously in Section 3.4.1. The absorbance of the shaken *E. coli* liquid was measured at 600 nm. After the LB medium was mixed, the turbidity became 0.145. The *E. coli* liquid and the ABA solution of each concentration (solvent: collagenase solution) were mixed at a ratio of 9:1. The mixtures were shaken at 100 rpm in a water bath of 37 °C for 24 h. An amount of 100 μL of each mixed solution was collected at 0 h, 1 h, and 3 h. Additionally, the absorbance was measured at 600 nm while the LB medium was a blank solution. 

#### 3.4.3. *E. coli* Culture on the Gel

The preparation method of the gels was as described in Section 3.3.1. The gelatin gels (Table 3) were taken out and poured into a 12-well plate. An amount of 0.05 M MES buffer was added at 2 mL/well drop by drop. The gels were then transferred to a 6-well plate, and 10 mL/well of 0.05 M MES buffer was dropped. After the supernatant was removed, 4 mL/well of 0.5% collagenase solution was added. Shaking was then carried out on a shaker under conditions of 200 rpm and 37 °C. After 0 h, 0.5 h, 1 h, 1.5 h, 2 h, and 24 h, 100 μL of the supernatant was collected and the absorbance of the ABA was measured at 300 nm using a blank. This solution was mixed with *E. coli* liquid at a ratio of 1:9. After mixing for 3 h, the turbidity was measured at 600 nm and calibrated to 100% transmittance with 0.5% collagenase solution. 

### 3.5. Antibacterial Activity of the Gel Surface 

#### 3.5.1. Observation of Double Zone Growth Inhibition

The preparation method of the gels was as described in Section 3.3.1. A phosphate-buffered saline (PBS) solution (−) was dropped at 1 mL/well, and the gels were left at 4 °C. *E. coli* liquid (turbidity 0.1) was diluted 10-fold and applied to the whole agar medium. Next, the liquid was rinsed with water and incubated at 37 °C. 

#### 3.5.2. *E. coli* Culture on the Gel

The preparation method of the gels was as described in Section 3.3.1. 1 mL/well of PBS (−) and 1 mL/well of LB medium were each added. The plate was allowed to stand at 4 °C, then 20 μL of *E. coli* liquid was dropped. After 1 week of culture, 1 mL of PBS (−) solution was added, the supernatants were removed, and the remaining liquid was diluted 1000-fold and applied to the agar medium. The number of colonies was measured after 1 day.

#### 3.5.3. Measurement of the Number of Bacteria in Rat Subcutaneous Implantation

The preparation method of the gels was as described in Section 3.3.1. PBS (−) solution was dropped and left at 4 °C. The gels were transplanted subcutaneously into nine-week-old rats. One week after the transplantation, the gels were excised and immersed in 1 mL of PBS solution. After 1 hour of PBS immersion and soaking, the gels were applied to the agar medium. The agar medium was cultured at 37 °C for 1 day, and the number of colonies was measured. 

#### 3.5.4. *E. coli* Bacterial Density in Relation to the Turbidity Method.

The preparation method of the gels was as described in Section 3.3.1. *E. coli* liquid was dropped at 2 mL/well. Turbidity was measured after culturing for 3 h under a 37 °C condition. The supernatant was removed, and again *E. coli* liquid was dropped. After 3 h, the turbidity was measured, and the process was repeated 7 times.

### 3.6. Evaluation of the Cytotoxicity of the Gel 

#### 3.6.1. HUVEC Culture on the Gel

The preparation method of the gels was as described in Section 3.3.1. Then, 1 mL of Dulbecco’s Modified Eagle Medium (DMEM) with fasting blood sugar (FBS 2%) was dropped and allowed to stand at 4 °C for 6 h. After the supernatant was removed, 1 mL of DMEM (FBS 2%) was dropped again and left to stand at 4 °C for 1 day. After that, 1 mL/well of EGM-2 was added and the same procedure was carried out. HUVECs were seeded at 5000 cells/well. The WST-8 assay was performed 4 days after seeding, and the medium exchange was carried out 1 day after sowing. An amount of 2 mL of medium and 300 μL of WST-8 solution were mixed, and the mixture was warmed at 37 °C for 5 minutes. Then, 230 μL/well of the mixed solution was dropped and incubated in an incubator for 4 h. Then, 30 μL of 0.1 N HCl was added. Thereafter, 100 μL of the supernatant was transferred to a 96-well plate. The absorbance was measured at 450 nm using a microplate reader. The measurement was performed on a Microplate Photometer from Thermo Scientific (Waltham, MA, USA).

#### 3.6.2. H&E Staining of the Gel after Subcutaneous Implantation in Rats 

An amount of 2 mL/well of MES buffer was added in each condition of Gelatin gel and ABA-Gelatin gel. After shaking overnight at room temperature, the supernatant suction was removed. Subcutaneous transplantation was performed in 9-week-old SD rats. All animal experiments were performed in accordance with the guidelines of the Ethics Committee on Animal Experiments and accepted by Kyushu University (A27-326-0, 19 Feb 2016). After 1 week, the tissue sample was taken out and immersed in 10% neutral buffered formalin solution. Tissue sections were cut out and stained with hematoxylin and eosin. 

## Figures and Tables

**Figure 1 gels-05-00032-f001:**
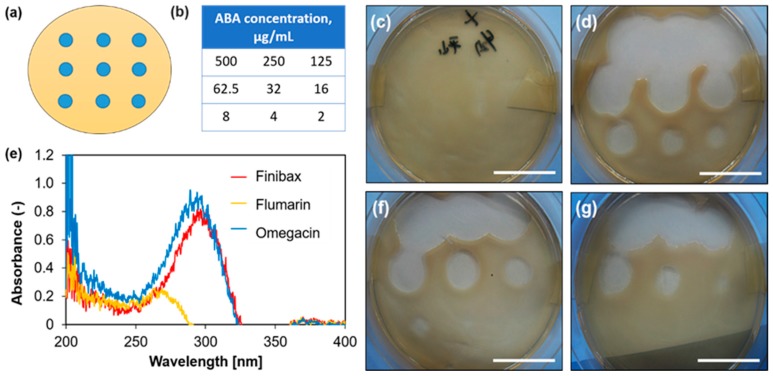
Descriptions of the images of the experiment: (**a**) The dropped position of the antibacterial agent (ABA) solutions in a dish; (**b**) the ABA concentrations: 2–500 µg/mL on the point; and (**e**) the spectral wavelength of the antibacterial agents, namely Finibax, Flomoxef, and Biapenem. Photographs of the action of the ABAs in a dish of transformed *Escherichia coli*: (**c**) control or ABA-free; (**d**) Finibax; (**f**) Flomoxef; and (**g**) Biapenem after 1 day of the inoculation. The bar represents 30 mm.

**Figure 2 gels-05-00032-f002:**
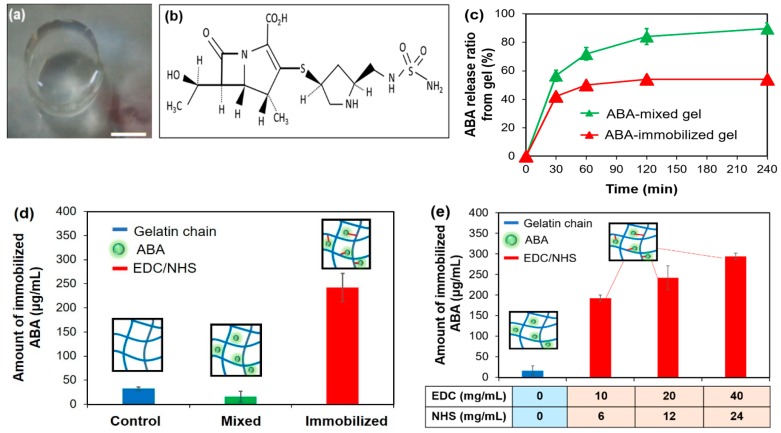
(**a**) Photograph of the ABA-Gelatin gel- the bar represents 5 mm; (**b**) the molecular structure of Finibax as an ABA; (**c**) the release percent of ABA in the gels; (**d**) the immobilized ABA amount in the ABA-free, ABA-mixed, and ABA-immobilized gelatins; and (**e**) the immobilized ABA amount at different concentrations of the crosslinking agent.

**Figure 3 gels-05-00032-f003:**
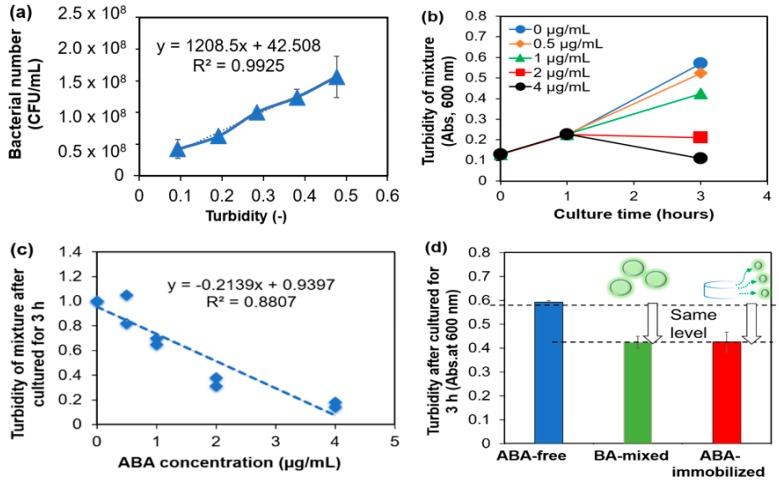
(**a**) Calibration curve of the number of *E. coli* (CFU/mL) and the turbidity value; (**b**) temporal change of the turbidity at different concentrations of the ABA; (**c**) calibration curve of the turbidity and the concentration of the ABA after three hours; and (**d**) evaluation of the activity of the ABA in free, mixed, and immobilized cases.

**Figure 4 gels-05-00032-f004:**
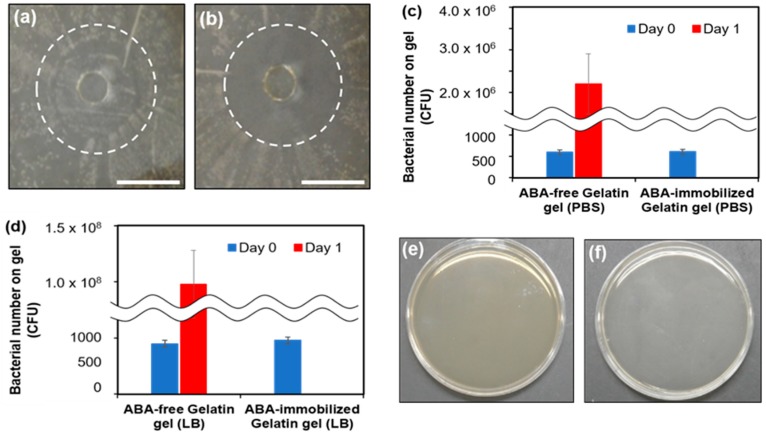
Photographs of the antibacterial evaluation on *E. coli*: (**a**) ABA-free gelatin gel and (**b**) ABA-immobilized gelatin gel. The bars represent 1 cm. Measurements of bacterial count on the gel surface: (**c**) PBS and (**d**) LB medium. Agar medium coated with PBS immersed in gel: (**e**) Gelatin without an antibacterial agent and (**f**) gelatin mixed with an antibacterial agent.

**Figure 5 gels-05-00032-f005:**
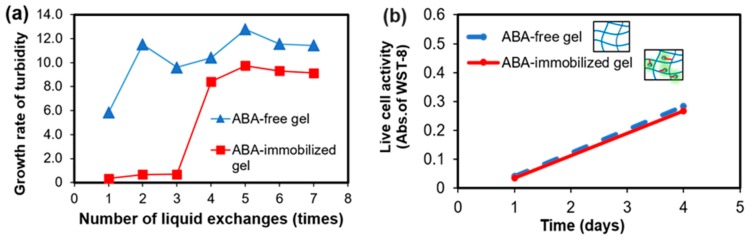
ABA-free gelatin gel and ABA-immobilized gelatin gel: (**a**) Turbidity increase rate with respect to the number of liquid changes and (**b**) evaluation of the number of viable cells in the human umbilical vein endothelial cells (HUVEC) culture on the gels.

**Figure 6 gels-05-00032-f006:**
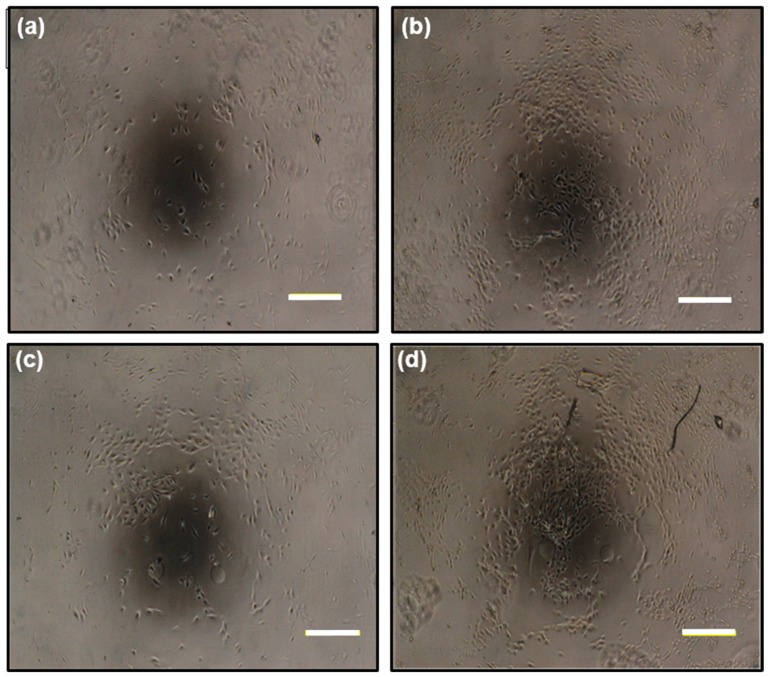
Cell photographs of the human umbilical vein endothelial cell (HUVEC) culture on gelatin gels: 1 day after seeding with (**a**) an ABA-free gelatin gel and (**b**) an ABA-immobilized gelatin gel; 4 days after seeding with (**c**) an ABA-free gelatin gel and (**d**) an ABA-immobilized gelatin gel. The bars represent 200 µm.

**Figure 7 gels-05-00032-f007:**
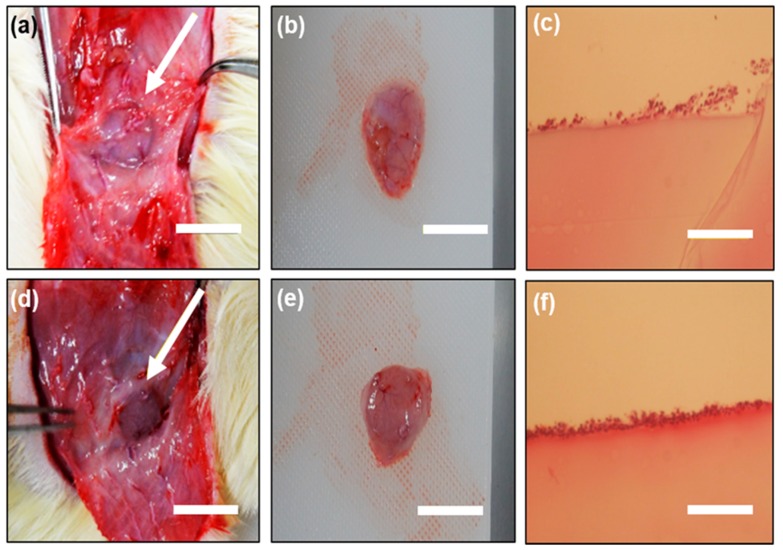
ABA-free gelatin gel: (**a**) 1 week after transplantation; (**b**) after the extraction of the gel and (**c**) hematoxylin and eosin (H&E) staining of the gel after subcutaneous implantation for 1 week in the rat; ABA-immobilized gelatin gel: (**d**) 1 week after transplantation; (**e**) after the extraction of the gel and (**f**) H&E staining of the gel after subcutaneous implantation for 1 week in the rat. Bars represent 1 cm and the arrows indicate the gel sample.

**Table 1 gels-05-00032-t001:** Antibacterial agents.

FDA *-Approved ABA	Trade Name	Category	Approval Year/Trial Year
Finibax	Doripenem	β-Lactam-carbapenem	2007
Flomoxef	Flumarin	β-Lactam-cephalosporins	1988
Biapenem	Omegacin	β-Lactam-carbapenem	2002

* US Food and Drug Administration, (FDA).

**Table 2 gels-05-00032-t002:** The molar ratio of the crosslinking couple.

Case No.	Concentration, mg/mL	
	EDC	NHS
1	0	0
2	10	6
3	20	12
4	40	24

**Table 3 gels-05-00032-t003:** Composition of the gels.

Condition Name	Composition
ABA-free	Collagenase solution mixed with *E. coli* liquid
Non-immobilized ABA	ABA solution (solvent: collagenase solution) mixed with *E. coli* liquid
Immobilized ABA	ABA-immobilized gel mixed with *E. coli* liquid

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
