# Peer review of "Antibacterial-Agent-Immobilized Gelatin Hydrogel as a 3D Scaffold for Natural and Bioengineered Tissues"

_gels, 2019, doi:10.3390/gels5020032_

Round 1

Reviewer 1 Report

Title: Antibacterial-agent-immobilized gelatin hydrogel as a 3D scaffold for natural and bioengineered tissues

This is a very interesting research and the aim of this study was to investigated hydrogel as an artificial bile duct and limited researchers worked in this research field using antibacterial agent within a hydrogel-based material.

Minor corrections:

-         Abstract: The final part which is results and conclusion of this study need to be expanded. Authors just provided very generic sentence at the end of abstract which is not enough.

-         Figure 1 – C,D, F and G are blared

-         - did you role the biomaterials to provide shape of duct?

-         This manuscript needs proper English proof reading as some sentences need to be re-phrased.

Overall, I accept this manuscript with minor correction.

Author Response

Response to Reviewer 1 comments

We would like to thank the editor and reviewers for their comments that helped us to improve our manuscript. The revision of this manuscript was based on reviewers’ comments and our responses are written in red as follows.

Reviewer 1

This is a very interesting research and the aim of this study was to investigated hydrogel as an artificial bile duct and limited researchers worked in this research field using antibacterial agent within a hydrogel-based material.

Minor corrections:

Point 1: Abstract: The final part which is the results and conclusion of this study need to be expanded. Authors just provided a very generic sentence at the end of abstract which is not enough.

Response 1: Thank you for bringing out this concern. Based on this comment, we added a more detailed explanation about the gel behaviors. Please see the line below.

lines: 28-30,

Overall, the ABA-Gelatin hydrogel was found to be viable for use in hydrogel applications for tissue engineering due to its good bactericidal ability, cell adhesion, and proliferation, as well as having no cytotoxicity to cells.

Point 2: Figure 1 – C,D, F and G are blared.

Response 2: Thank you for this comment and we apologize for the unclear view regarding this statement. In the manuscript, we modified these figures. Please see the lines below.  

Figure 1.

 Point 3: This manuscript needs proper English proofreading as some sentences need to be re-phrased.

Response 3: The manuscript has been certified English editing service (ID: English-9619) by MDPI. However, after receiving this comment, we also sent the manuscript to proofreading. They have reviewed this case more. Thank you so much for this comment.   

Reviewer 2 Report

In this manuscript, the authors conjugated antibacterial agent onto gelatin hydrogel to reduce the infection risk for the potential application as artificial bile duct. The hydrogel exhibited good antibacterial property and good biocompatibility, showing promise for its application in tissue engineering. Overall, the authors have done an interesting and in-depth research with well arranged experiments and got some meaningful conclusions which would benefit the research of other scientists in this field. However, some minor problems need to be addressed before publication.

1.     Figure 1C, D, F, G, the position of the dropped position should be marked and the scale of the pictures should be the same for easier comparison (the plate in picture G looks bigger). In the legend, A, B,E, C, D, F, G should be used instead of a, b, e, c, d, f, g for matching the labels in the pictures. The same change should be made for other figures. And they had better follow the alphabetical sequence.

2.     Figure 2, there are some numbers on picture A. Figure 4, there are numbers close to picture G. Figure 5, there is a number under picture B.

3.     Figure 4, D and E, the author should put something for the bacterial number on gelatin for easier comparison for the readers instead of leaving it blank.

Author Response

Response to Reviewer 2 comments

We would like to thank the editor and reviewers for their comments that helped us to improve our manuscript. The revision of this manuscript was based on reviewers’ comments and our responses are written in red as follows.

Reviewer 2

Comments and Suggestions for Authors

In this manuscript, the authors conjugated antibacterial agent onto gelatin hydrogel to reduce the infection risk for the potential application as artificial bile duct. The hydrogel exhibited good antibacterial property and good biocompatibility, showing promise for its application in tissue engineering. Overall, the authors have done an interesting and in-depth research with well arranged experiments and got some meaningful conclusions which would benefit the research of other scientists in this field.

However, some minor problems need to be addressed before publication.

Point 1: Figure 1C, D, F, G, the position of the dropped position should be marked and the scale of the pictures should be the same for easier comparison (the plate in picture G looks bigger). In the legend, A, B,E, C, D, F, G should be used instead of a, b, e, c, d, f, g for matching the labels in the pictures. The same change should be made for other figures. And they had better follow the alphabetical sequence.

Response 1: We very appreciate your comments and thank you sincerely. According to your comments, we modified these figures.

Lines: Figure 1.

Point 2: Figure 2, there are some numbers on picture A. Figure 4, there are numbers close to picture G. Figure 5, and there is a number under picture B.

Response 2: We are glad that you highlighted such important detail and we clearly apologize for this mishap.

Point 3: Figure 4, D and E, the author should put something for the bacterial number on gelatin for easier comparison for the readers instead of leaving it blank.

Response 3: We modified the figure and also mentioned bacterial number in the illustration section.

Lines: 223- 225

Bacterial number on the gel surface that increased from 539 (on day 0) to 2.32 x 106 in PBS and 1.01 x 108 in LB medium (on day 1).
